# Fungal Melanonychia: A Systematic Review

**DOI:** 10.3390/microorganisms12061096

**Published:** 2024-05-28

**Authors:** Carmen Rodríguez-Cerdeira, Erick Martínez-Herrera, Paulina Nundehui Cortés-López, Estefanía Guzmán-Montijo, Carlos Daniel Sánchez-Cárdenas, Roberto Arenas, Claudia Erika Fuentes-Venado, Diana Carolina Vega-Sánchez, Rodolfo Pinto-Almazán

**Affiliations:** 1Dermatology Department, Hospital Vithas Vigo, Vía del Norte 48, 36206 Vigo, Spain; 2Department of Health Sciences, University of Vigo, Campus of Vigo, As Lagoas, 36310 Vigo, Spain; 3Fundación Vithas, Grupo Hospitalario Vithas, 28043 Madrid, Spain; eomartinez@ipn.mx (E.M.-H.); jefegrillo@gmail.com (C.D.S.-C.); rarenas98@hotmail.com (R.A.); 4Ibero-Latin American College of Dermatology (CILAD), Buenos Aires C1091, Argentina; 5Sección de Estudios de Posgrado e Investigación, Escuela Superior de Medicina, Instituto Politécnico Nacional, Plan de San Luis y Díaz Mirón, Ciudad de México 11340, Mexico; cefvenado@hotmail.com; 6Sección de Micología, Hospital General “Dr. Manuel Gea González”, Tlalpan, Ciudad de México 14080, Mexico; pauncortes@gmail.com (P.N.C.-L.); estefy2998@gmail.com (E.G.-M.); dianavss84@gmail.com (D.C.V.-S.); 7Facultad de Medicina, Universidad Nacional Autónoma de México, Coyoacán, Ciudad de México 04360, Mexico; 8Servicio de Dermatología, Centro Médico Nacional La Raza, Azcapotzalco, Ciudad de México 04360, Mexico; 9Servicio de Medicina Física y Rehabilitación, Hospital General de Zona No 197, Texcoco 56108, Mexico

**Keywords:** fungal melanonychia, *Trichophyton rubrum*, *Neoscytalidium dimidiatum*, dermatoscopy, melanin

## Abstract

Fungal melanonychia is an uncommon condition, most typically caused by opportunistic melanin-producing pigmented filamentous fungi in the nail plate. In the present study, the clinical characteristics of patients diagnosed with fungal melanonychia were analyzed through a systematic review of cases reported in the literature. The MESH terms used for the search were “melanonychia” AND “fungal” OR “fungi” through four databases: PubMed, SciELO, Google scholar and SCOPUS. After discarding inadequate articles using the exclusion criteria, 33 articles with 133 cases were analyzed, of which 44% were women, 56% were men and the age range was between 9 and 87 years. The majority of cases were reported in Turkey followed by Korea and Italy. Frequent causal agents detected were *Trichophyton rubrum* as non-dematiaceous in 55% *and Neoscytalidium dimidiatum* as dematiaceous in 8%. Predisposing factors included nail trauma, migration history, employment and/or outdoor activities. Involvement in a single nail was presented in 45% of the cases, while more than one affected nail was identified in 21%, with a range of 2 to 10 nails. Regarding the clinical classification, 41% evidenced more than one type of melanonychia, 21% corresponded to the longitudinal pattern and 13% was of total diffuse type. Likewise, the usual dermoscopic pattern was multicolor pigmentation. It is concluded that fungal melanonychia is an uncommon variant of onychomycosis and the differential diagnosis is broad, which highlights the complexity of this disease.

## 1. Introduction

Melanonychia is the term used to describe the presence of melanin in the nail plate or melanocytes within the matrix (*Melanocytic hyperplasia*), which is observed as dark brown pigmentation, which may or may not be diffused [1,2,3,4]. Depending on its topography, different clinical types have been described: longitudinal pattern, linear distal, diffuse proximal or distal, and diffuse total [5]. It can be located in the fingernails or toenails, usually affecting only one nail. This onychopathy is more prevalent in adults than in children, with men being the most affected (60.6%). The risk of acquiring the disease increases with age after 50 [1,2,3,4,5].

Fungal melanonychia is an uncommon condition, and it has the most frequent association with distal subungual followed by the total dystrophic form [1]. Nail disorders due to cutaneous or general diseases with a similar alteration in the nail configuration may be confused with onychomycosis. The differential diagnosis includes subungual melanoma, subungual hemorrhage, drug-induced pigmentation, endocrine diseases, and nevi [6]. In their meta-analysis, Lim et al. demonstrated that some characteristics are commonly identified through dermoscopy that allow for the diagnosis of onychomycosis [7]. On the other hand, as in other mycoses, the accurate diagnosis of nail mycosis is essential since systemic treatments are necessary for long periods of time (2–3 months), including topical treatments for more than a year. For diagnosis, microscopic examination and culture and/or molecular testing (nail scrapings) are necessary to limit misdiagnoses and provide adequate treatments [7,8].

Until now, 21 species causing fungal melanonychia have been reported, mostly caused by opportunistic pigmented filamentous fungi that produce melanin pigments. These pigments are incorporated into the cell wall or excreted extracellularly, which is why these fungi are called phaeoid or dematiaceous. Generally, these mycoses are caused by species such as *Neoscytalidium dimidiatum* (previously *Scytalidium dimidiatum*), *Exophiala* spp. and *Alternaria* spp. [5,6]; although, it can also be caused by non-dematiaceous fungi, such as dermatophytes like *Trichophyton rubrum*. The *Candida* genus frequently causes color changes from yellow to green or black; cases have been described due to *C. albicans*, *C. parapsilosis*, *C. humicula* and *C. tropicalis* [1,5,6,7].

It is suggested that the melanonychia produced by these agents is due to the activation of host melanocytes by the inflammation [4]. The melanin they produce works as an armor that protects them from the environment and ultraviolet light, among other external factors. Some strains can produce a diffusible black pigment, *Aspergillus* spp. colors brown by producing DOPA melanin through the enzymes tyrosinase and laccase [8,9], and *Candida* spp. metabolizes d-tryptophan into pigmented products [10] and produces DOPA melanin through laccase [11,12].

Diagnosis is made by direct examination with potassium hydroxide or chlorazol black E culture, nail unit biopsy with periodic acid-Schiff (PAS) stain, and polymerase chain reaction. Among the dermoscopic findings are homogeneous brown pigmentation (35.7%) and homogeneous gray and black color (21.4%) [1]. Other findings are multicolor pigmentation, black pigmented aggregates, superficial transverse striation and blurred appearance [13].

In recent years, an increase in the number of cases of fungal melanonychia has been observed along with the rise in the etiological agents involved. This study aims to provide information on demographics, clinical data, dermoscopic patterns, diagnostic methods, etiological agents and treatment effectiveness.

## 2. Methodology

An advanced search was performed in English and Spanish through the databases: Medical Literature Analysis and Retrieval System Online (MEDLINE/PUBMED), Scientific Electronic Library Online (SciELO), Google scholar and SCOPUS, looking for case reports, observational studies and clinical trials, from March 1992 to May 2023.

The used terms were “melanonychia” AND “fungal” OR “fungi”. The total number of articles found was 253, and the review was carried out based on Preferred Reporting Items for Systematic reviews and Meta-Analyses (PRISMA) (See Figure 1).

To refine for relevant studies, the search criteria included case reports, case series, observational studies and clinical trials. After reading the titles and analyzing the full text, the most relevant articles for the review that met the inclusion and exclusion criteria were chosen. At the end of the process, 33 original articles were chosen; however, the review was limited to articles that offered detailed descriptions of fungal melanonychia as well as diagnostic methods and treatment.

Studies that did not detail the presence of fungal melanonychia; studies where melanonychia was caused by other diseases; as well as conference posters, reviews, meta-analyses, systematic reviews or articles that did not include case reports were excluded. Two independent reviewers (P.N.C.-L. and E.G.-M) evaluated the titles, abstracts and full texts of each potential study. (E.M.-H. and R.P.-A.) resolved any details regarding study inclusion and evaluated methodological quality.

In order to ensure data accuracy, duplicate articles were excluded. Collected data included country; type of study; demographic data; and clinical manifestations such as topography, dermoscopy, use of diagnostic methods, administered treatments and outcomes.

Regarding the identification of species, the reported taxonomy from studies was used; although, it is recognized that a new taxonomy is currently applied to name the various species. To describe melanonychia patterns, the classifications by the authors from the original papers were also used. However, when reviewing the scientific articles, a reclassification of melanonychias was carried out when there were photos that involved the classifications mentioned by Starace et al. [14] and Kim et al. [15].

For the quality of risk of bias, an analysis was carried out in duplicate (E.M.-H. and R.P.-A.) using the JBI Critical Appraisal Checklist for Systematic Reviews and Research Syntheses and Critical Appraisal Skills Program (CASP) tools, which serve to analyze the quality and risk of bias of qualitative systematic reviews.

## 3. Results

Of the 33 selected articles, 133 patients who met the inclusion criteria were taken into account (Table 1 and Table 2). The three countries with the highest number of reported cases were Turkey 36% (n = 48), Korea 27% (n = 36) and Italy 16% (n = 21). In terms of total cases, 44% (n = 59) of them were women and 56% (n = 74) represented men. The mean age of presentation was 45.4 years in a range of 9 to 87 years [14,15,16,17,18,19,20,21,22,23,24,25,26,27,28,29,30,31,32,33,34,35,36,37,38,39,40,41,42,43,44,45,46].

In terms of occupations, 95% (n = 126) did not have such information, the remaining 5% (n = 7) corresponded to green tea leaf pickers 1.5% (n = 2), activities related to gardening 2% (n = 3), farmers 0.75% (n = 1) and magistrates 0.75% (n = 1) [14,15,16,17,18,19,20,21,22,23,24,25,26,27,28,29,30,31,32,33,34,35,36,37,38,39,40,41,42,43,44,45,46].

The following background were identified: 11.3% (n = 15) of the cases were related to migration 3.8% (n = 5), cancer 3% (n = 4), nail trauma 1.5% (n = 2), antifungals 0.8% (n = 1), ethnic pigmentation 0.8% (n = 1), longitudinal melanonychia 0.8% (n = 1) and outdoor activity 0.8% (n = 1) [14,15,16,17,18,19,20,21,22,23,24,25,26,27,28,29,30,31,32,33,34,35,36,37,38,39,40,41,42,43,44,45,46].

Likewise, present comorbidities included systemic diseases 26% (n = 34), mycosis 2% (n = 3), cancer 2% (n = 3), immunosuppression states 2% (n = 3) and others 2% (n = 2); 16% (n = 21) reported themselves as healthy; and in 50% (n = 67), pertinent information regarding medical history was not provided. The average evolution time was 1.71 years with a range of 0.08 to 10.6 years [14,15,16,17,18,19,20,21,22,23,24,25,26,27,28,29,30,31,32,33,34,35,36,37,38,39,40,41,42,43,44,45,46].

In total, 171 affected nails were recorded. A trend was observed towards the toenails 62% (n = 82), especially in the first nail 43% (n = 57). Furthermore, the nails at the level of the hands had an involvement of 24% (n = 32), whereas fingernails and toenails were both compromised in 2% (n = 3); in 12% of the cases (n = 16), no information on the location of affected nails was provided. The average number of nails involved was 8 [14,15,16,17,18,19,20,21,22,23,24,25,26,27,28,29,30,31,32,33,34,35,36,37,38,39,40,41,42,43,44,45,46].

Regarding the classification of melanonychia by case, 41% (n = 55) presented more than one type, 21% (n = 28) were longitudinal, 13% (n = 18) total diffuse, 10% (n = 13) distal diffuse, 2% (n = 2) proximal diffuse, 5% (n = 6) distal linear and in 8% (n = 11) not enough information was obtained to classify the melanonychia pattern [14,15,16,17,18,19,20,21,22,23,24,25,26,27,28,29,30,31,32,33,34,35,36,37,38,39,40,41,42,43,44,45,46].

For diagnostic methods, direct examination was used in 92% (n = 122), culture in 59% (n = 79), biopsy in 35% (n = 47), dermoscopy in 80% (n = 106), molecular biology was performed in 5% (n = 6) and serological tests were applied in 1% (n = 1). Within the direct examination, dematiaceous hyphae were found in 4% (n = 5), hyaline hyphae or spores in 10% (n = 13) and not specified in 78% (n = 104) [14,15,16,17,18,19,20,21,22,23,24,25,26,27,28,29,30,31,32,33,34,35,36,37,38,39,40,41,42,43,44,45,46].

A total of 149 nails were analyzed with dermatoscopy; of these, the most reported findings were 31% (n = 46) nails with multicolor pigmentation, 22% (n = 33) yellow–white spots, 19% (n = 28) superficial white scale, 18%(n = 27) hyperkeratosis, 15% (n = 22) inverse triangle pattern, 14% (n = 21) superficial transverse striation, 14% (n = 21) pigmentation with irregular borders, 13% (n = 20) longitudinal striae, 11% (n = 17) irregular matte pigmentation with longitudinal striae, 5% (n = 8) pseudo-Hutchinson, and in 1% (n = 2) some characteristics such as roughness, onycholysis and aurora pattern borealis were observed [14,15,16,17,18,19,20,21,22,23,24,25,26,27,28,29,30,31,32,33,34,35,36,37,38,39,40,41,42,43,44,45,46].

The causal agents were identified in 53% (n = 71) of the cases. These corresponded to *T. rubrum* 55% (n = 39), *S. dimidiatum* 8% (n = 6), *C. parapsilosis* 7% (n = 5), *C. albicans* 3% (n = 3), *C. tropicalis* 3% (n = 2), *Candida* spp. 3% (n = 2), *Aspergillus niger* 3% (n = 2), *Fonsecaea pedrosoi* 1% (n = 1), *Fusarium solani* 1% (n = 1), *F. oxysporum* 1% (n = 1), *Exophiala dermatitidis* 1% (n = 1), *Wangiella dermatitidis* 1% (n = 1), *C. tropicalis* and *A. niger* 1% (n = 1), *Botryosphaeria dothidea* 1% (n = 1), *Cladosporium* spp. 1% (n = 1), *T. interdigitale* 1% (n = 1), *A. alternata* 1% (n = 1), *Scytalidium* spp. 1% (n = 1) and *Sporothrix* spp. 1% (n = 1) [14,15,16,17,18,19,20,21,22,23,24,25,26,27,28,29,30,31,32,33,34,35,36,37,38,39,40,41,42,43,44,45,46].

Susceptibility testing was performed in only three cases, where *Candida parapsilosis* was found susceptible to ravuconazole, itraconazole, voriconazole, amphotericin B and micafungin. *B. dothidea* resulted as susceptible to amphotericin B, micafungin, terbinafine and voriconazole, and *W. dermatitidis* was susceptible to amphotericin B, 5-fluocytocin and ketoconazole [14,15,16,17,18,19,20,21,22,23,24,25,26,27,28,29,30,31,32,33,34,35,36,37,38,39,40,41,42,43,44,45,46].

Concerning the treatment, it was only described in 28% (n = 38). Itraconazole monotherapy was used in 7% (n = 9); terbinafine in 2% (n = 3); unspecified systemic antifungal in 2% (n = 3); amorolfine in 2% (n = 3); fluconazole in 1.5% (n = 2); fosravuconazole in 0.7% (n = 1); griseofulvin in 0.7% (n = 1); bifonazole in 0.7% (n = 1); not specified topical in 0.7% (n = 1); topical eficonazole in 0.7% (n = 1); combination of itraconazole and ciclopirox in 3% (n = 4); itraconazole and amorolfine in 0.7% (n = 1); Castellani formula and clotrimazole in 0.7% (n = 1); avulsion and itraconazole in 0.7% (n = 1); ciprofloxacin and itraconazole in 0.7% (n = 1); terbinafine and itraconazole in 0.7% (n = 1); itraconazole and eficonazole in 0.7% (n = 1); urea and terbinafine in 0.7% (n = 1); and terbinafine, bifonazole and urea in 0.7% (n = 1). In 72% (n = 96), no treatment was described [14,15,16,17,18,19,20,21,22,23,24,25,26,27,28,29,30,31,32,33,34,35,36,37,38,39,40,41,42,43,44,45,46].

As for results, there was no clinical report in 72% of the cases (n = 96) after the application of the antifungal treatment because this information was not mentioned in the studies (14–46). On the other hand, for those in which the outcome was specified 28% (n = 38), we highlight that in 18% of the cases (n = 24) a clinical improvement was obtained after starting treatment, and in 9% (n = 12) they experienced both a clinical and mycological cure. Only 1% (n = 1) experienced therapeutic failure [14,15,16,17,18,19,20,21,22,23,24,25,26,27,28,29,30,31,32,33,34,35,36,37,38,39,40,41,42,43,44,45,46].

## 4. Discussion

Fungal melanonychia is a particular variant of onychomycosis caused by non-dermatophyte yeasts and molds that have, as a common characteristic, the production of melanin or melanin-like pigments. There has been an increase in the incidence of onychomycosis caused by non-dermatophyte molds, including dematiaceous molds [13,20].

With respect to epidemiology, fungal melanonychia is more common in men and its prevalence increases with age [44]. This premise is consistent with this study since it was observed that 56% (n = 74) of the cases were men and the average age was 45.4 years. According to pre-existing evidence [20], in this study, the most affected nail was the first toenail [14,15,16,17,18,19,20,21,22,23,24,25,26,27,28,29,30,31,32,33,34,35,36,37,38,39,40,41,42,43,44,45,46].

Regarding the number of affected nails, in 45% (n = 60) only one nail was affected, while in 21% (n = 28) more than one affected nail was identified with a range of 2 to 10. This pattern suggests that there is self-inoculation by the patient [14,15,16,17,18,19,20,21,22,23,24,25,26,27,28,29,30,31,32,33,34,35,36,37,38,39,40,41,42,43,44,45,46].

This condition can manifest itself in both immunosuppressed and immunocompetent hosts [13]. Some of the following have been identified as risk factors: nail trauma; contact with wood or soil; and residence or migration to endemic areas close to the Equator, such as Africa, India, Thailand and the Caribbean [45]. This study identified that 8% (n = 11) of the cases had a history of migration or were involved in jobs and/or outdoor activities, including work related to gardening. In addition, in only 1.5% (n = 2) of the cases, collecting green tea leaves in India was documented as a profession, and in 1.5% (n = 2) previous nail trauma was documented [14,15,16,17,18,19,20,21,22,23,24,25,26,27,28,29,30,31,32,33,34,35,36,37,38,39,40,41,42,43,44,45,46].

With reference to etiology, the most isolated dematiaceous fungus was *N. dimidiatum* 8% (n = 6) along with one case of *Neoscytalidium* spp. (n = 1), and the most frequent non-dematiaceous dermatophyte was *T. rubrum* 55% (n = 39). Only in one case was the growth of two species recorded in the culture, which were *C. tropicalis* and *A. niger* [14,15,16,17,18,19,20,21,22,23,24,25,26,27,28,29,30,31,32,33,34,35,36,37,38,39,40,41,42,43,44,45,46].

In agreement with Finch et al., from a review of fungal melanonychia carried out in 2012, they described 21 species of dematiaceous and eight species of non-dematiaceous in this disease. They also reported that the most isolated dematiaceous was *N. dimidiatum* and the most common non-dematiaceous was *T. rubrum*, followed by dematiaceous fungi of the *Alternaria* and *Exophiala* spp. [46]. However, it is acknowledged that in this review, in 10% (n = 13) of the cases, various species of *Candida* were identified as the causal agent, as well as fungi that had not been previously reported in the literature such as *Sporothrix* spp. and *B. dothidea*. This finding underscores the relevance of expanding the information on causal agents of fungal melanonychia [14,15,16,17,18,19,20,21,22,23,24,25,26,27,28,29,30,31,32,33,34,35,36,37,38,39,40,41,42,43,44,45,46].

The pigments produced by fungi perform various functions. Firstly, they have the ability to act as powerful scavengers of free radicals released by macrophages and neutrophils, favor the virulence of the fungus and allow evasion of the host’s immune system. In addition, they have the ability to sequester some drugs, antifungal enzymes and antimicrobial peptides [12,20].

Some of the pigments produced by these fungi are melanins; of these, the predominant ones are of the dihydroxyphenylalanine (DOPA) and dihydroxynaphthalene (DHN) types [47,48]. Besides these common fungal melanins, there is evidence that some fungi produce pyomelanins as a result of the breakdown of aromatic amino acids, particularly tyrosine [12].

The diagnosis of this disease requires a combination of clinical and diagnostic methods. The most distinctive clinical feature is the presence of brown to black discoloration of the nail plate. In addition to discoloration, other clinical manifestations include dystrophy, onycholysis, thickening, subungual hyperkeratosis and paronychia [13].

The clinical pattern of nail involvement can provide clues to the origin of the infection. In *Neoscytalidium* species, affected nails are severely thickened and keratin debris may accumulate under the nail. Distal subungual onychomycosis and occasionally distal onycholysis are the clinical patterns of *Alternaria* nail infection, while longitudinal melanonychia is more common in dermatophyte strains such as *T. rubrum* [13,46,49,50,51,52,53]. However, most cases of melanonychia are difficult to distinguish because their clinical presentations are similar [54,55,56].

The clinical classification of melanonychias has divided them into a longitudinal, distal linear, proximal or distal diffuse and total diffuse form [20,32]. However, this terminology is not internationally approved yet.

In the present study, 41% (n = 55) of the cases presented more than one type of melanonychia in various nails, 21% (n = 28) corresponded to the longitudinal pattern and 13% (n = 18) to the total diffuse form. In 2021, Starace et al. described the clinical patterns of 20 patients and a total of 48 pigmented nails and found that the nail lesions affected the feet in a higher percentage of cases, with the first toe being the most affected—a finding that coincides with the results of this review. In accordance with this, longitudinal melanonychia was identified in 20 cases: eight of partial diffuse distal type, two diffuse partial proximal, one linear distal and 17 diffuse total [20].

In this onychopathy, mycological examinations take on vital importance. The direct examination with KOH is positive in 95% of cases [44], while the culture constitutes a tool that allows the microscopic characteristics of the fungus. However, its limitation is that it does not always provide positive fungal isolation; therefore, other complementary diagnostic techniques are used such as dermoscopy or molecular biology [46,50]. In the present study, in 92% (n = 122) of the cases, KOH test was used as one of the main diagnoses [14,15,16,17,18,19,20,21,22,23,24,25,26,27,28,29,30,31,32,33,34,35,36,37,38,39,40,41,42,43,44,45,46].

Regarding dermoscopy, continuous finding in the present review was multicolor pigmentation in 31% (n = 46). Starace et al. frequently reported the presence of multicolor pigmentation in 31.25%, irregular matte pigmentation in longitudinal striae in 31.25%, superficial transverse striation in 47.9%, scales on the surface of the nail in 45.8%, white or yellow strikes in 37.5% and jagged edges in 29.1% [4]. In 2020, an article from Hak-Jun et al. reported that multicolor pigmentation was present in 65% of cases [32].

The results obtained about diagnostic methods indicate that the approach to fungal melanonychia cannot depend exclusively on clinical evaluation; complementary tools such as direct examination, culture and dermoscopy must be used.

Currently, there are no established practice guidelines to address fungal melanonychia. According to the findings of this study, since fungal melanonychia is a disease that does not compromise life, case reports are not made including the treatments used or their effectiveness [57]. In the cases in which the treatment and outcome were detailed, it was observed that 95% (n = 36) obtained a clinical improvement; of these, 63% (n = 24) reported gradual clinical progress. On the other hand, 32% (n = 12) obtained a clinical and mycological cure. The most common therapeutic modality was the use of azole regimens, either in monotherapy or in combination, with itraconazole being the most used antifungal [14,15,16,17,18,19,20,21,22,23,24,25,26,27,28,29,30,31,32,33,34,35,36,37,38,39,40,41,42,43,44,45,46].

## 5. Conclusions

Fungal melanonychia is an uncommon variant of onychomycosis, the differential diagnosis is broad, which highlights the complexity of this disease. We underscore the diversity of causal agents, some of which are reported as unique cases in the literature.

The prevalence in men is notable; however, the lack of information on the occupation of the cases limits a more detailed analysis. The clinical classification and dermoscopic findings underline the need for homogenization in this onychopathy. With regard to the etiology, *T. rubrum* and *N. dimidiatum* were observed to be the most frequent causal agents. Due to this, diagnosis is necessary to guide appropriate treatment corresponding to the etiology.

## Figures and Tables

**Figure 1 microorganisms-12-01096-f001:**
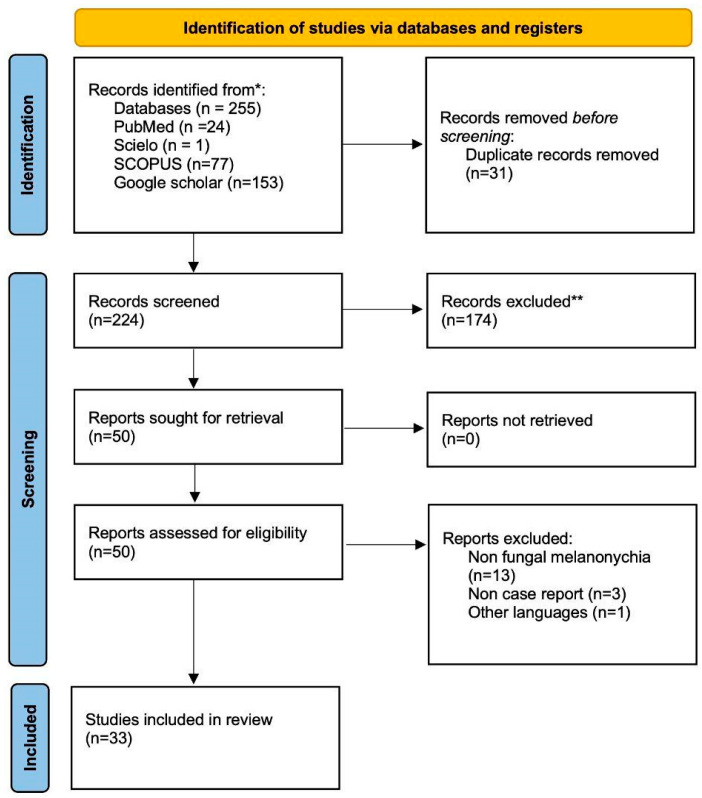
Flowchart of the different phases of the systematic review. * 4 Bases were analyzed: Pubmed, Scopus, Scielo, Google Scholar ** Articles exclude because: studies where melanonychia was caused by other diseases; as well as conference posters, reviews, meta-analyses, systematic reviews or articles that did not include case reports.

**Table 1 microorganisms-12-01096-t001:** Epidemiology of fungal melanonychia.

Country	Age	Sex	Onset (Years)	Topography																				Melanonychia Pattern	Diagnostic Method	Dermoscopy	Etiological Agent	Treatment	Outcome	Reference
				Right Hand					Left Hand					Right Foot					Left Foot											
				1	2	3	4	5	1	2	3	4	5	1	2	3	4	5	1	2	3	4	5							
Belgium	36	M	-	-																				-	B and C	-	*S. dimidiatum*	FLC	Clinical cure	[16]
	41	M	-	-																				-	B and C			ITC	Clinical cure	
	54	M	-	-																				-	B and C			TRB	Clinical cure	
	58	M	-	-																				-	B and C			TRB	Clinical Cure	
France	33	F	0.25																					Distal linear	B and C	-	*T. rubrum*	-	-	[17]
	65	M	-																					Distal diffuse	B and C	-	*T. rubrum*	-	-	
	60	M	1																					Longitudinal	DE, B and C	-	*C. parapsilosis*	AMO 5%	Cure	[18]
India	40	F	4																					Total diffuse	DE and C	-	*S. dimidiatum*	-	-	[19]
	35	F	2	Thumbnail																				-	DE and C	-	*S. dimidiatum*	-	-	
	25	F	1	-																				Proximal diffuse	B, C and D	Hyperpigmentation, scaling, yellow streaks, *aurora borealis*	*T. rubrum*	ITC, Ciclopirox	Improve	[20]
	39	M	0.16	-																				Proximal diffuse	B, C and D		*T. rubrum*	ITC, Ciclopirox	Improve	
	36	F	0.75	-																				Longitudinal	B, C and D		*T. rubrum*	ITC, Ciclopirox	Improve	
	45	F	0.75	-																				Longitudinal	B, C and D		*T. rubrum*	ITC, Ciclopirox	Improve	
Italy	65	F	1																					Total diffuse	DE and C	-	*E. dermatitidis*	AMO 5%	Improve	[21]
	43	F	-																					Total diffuse	DE, C and D	Grey–black pigmentation, superficial transverse striation, scaling	*T. rubrum*	-	-	[14]
	16	M	-																					Total diffuse	DE, C and D	Grey-black pigmentation	*T. rubrum*	-	-	
	54	F	-																					Diffuse	DE, C and D	Multicolored pigmentation, jagged edges	*T. rubrum*	-	-	
	66	M	-																					Diffuse	DE, C and D	Irregular pigmentation in longitudinal striae	*T. rubrum*	-	-	
	33	M	-																					Longitudinal	DE, C and D	Reverse triangle, irregular pigmentation in longitudinal striae	*T. rubrum*	-	-	
	66	F	-																					Diffuse, longitudinal	DE, C and D	Multicolored pigmentation, superficial transverse striation, jagged edges, yellow streaks, subungual hyperkeratosis	*T. rubrum*	-	-	
	73	M	-																					Diffuse, longitudinal	DE, C and D	Yellow streaks, subungual hyperkeratosis, scaling, jagged edges, transverse striation, multicolored pigmentation, reverse triangle	*T. rubrum*	-	-	
	45	F	-																					Longitudinal	DE, C and D	Reverse triangle, jagged edges, yellow streaks, superficial transverse striation, scaling, subungual hyperkeratosis	*T. rubrum*	-	-	
	82	M	-																					Total diffuse	DE, C and D	Subungual hyperkeratosis, scaling, yellow streaks, multicolored pigmentation	*T. rubrum*	-	-	
	34	F	-																					Longitudinal	DE, C and D	Yellow streaks, reverse triangle, jagged edges, multicolored pigmentation, irregular pigmentation in longitudinal striae	*T. interdigitale*	-	-	
	73	M	-																					Diffuse, longitudinal	DE, C and D	Subungual hyperkeratosis, scaling, yellow streaks, superficial transverse striation, irregular pigmentation in longitudinal striae	*T. rubrum*	-	-	
	36	F	-																					Longitudinal	DE, C and D	Subungual hyperkeratosis, scaling, yellow streaks, reverse triangle, superficial transverse striation	*T. rubrum*	-	-	
	72	M	-																					Total diffuse, longitudinal	DE, C and D	Subungual hyperkeratosis, scaling, yellow streaks, superficial transverse striation, irregular pigmentation in longitudinal striae	*T. rubrum*	-	-	
	71	M	-																					Total diffuse longitudinal	DE, C and D	Multicolored pigmentation, subungual hyperkeratosis	*T. rubrum*	-	-	
	31	F	-																					Distal partial diffuse	DE, C and D	Subungual hyperkeratosis, scaling, yellow streaks, jagged edges, superficial transverse striation, irregular pigmentation in longitudinal striae	*T. rubrum*	-	-	
	67	M	-																					Distal partial diffuse	DE, C and D	Subungual hyperkeratosis, yellow streaks, jagged edges, superficial transverse striation, irregular pigmentation in longitudinal striae	*T. rubrum*	-	-	
	28	M	-																					Total diffuse	DE, C and D	Subungual hyperkeratosis, scaling, yellow streaks, irregular pigmentation in longitudinal striae	*T. rubrum*	-	-	
	58	F	-																					Distal partial diffuse	DE, C and D	Subungual hyperkeratosis, scaling, yellow streaks, jagged edges, superficial transverse striation, irregular pigmentation in longitudinal striae	*T. rubrum*	-	-	
	46	M	-																					Longitudinal	DE, C and D	Subungual hyperkeratosis, jagged edges, superficial transverse striation, irregular pigmentation in longitudinal striae	*T. rubrum*	-	-	
	21	F	-																					Longitudinal	DE, C and D	Subungual hyperkeratosis, yellow streaks, jagged edges, reverse triangle, irregular pigmentation in longitudinal striae	*T. rubrum*	-	-	
Japan	73	F	6																					Total diffuse	DE, B, C, D and PCR	Multicolored pigmentation	*C. parapsilosis*	RVZ	Improve	[22]
	29	F	-																					Total diffuse	DE	-	*C. parapsilosis*	GRIS	Failed	
	82	M	0.5																					Proximal diffuse	DE, B, C, D and PCR	Scaling, irregular black dots	*B. dothidea*	EFZ 10%	Cured	[23]
Korea	64	M	4																					-	DE, C	-	*F. solani*	ITC	Improve	[24]
	65	F	1																					Diffuse	DE, B and C	-	*C. albicans*	ITC, AMO	Improve	[25]
	48	M	2																					Total diffuse	B, C, D and PCR	Multicolored pigmentation, superficial transverse striation	*T. rubrum*	ITC	Clinical cure	[26]
	49	F	1																					Longitudinal	B, C and PCR	-	*F. oxysporum*	ITC	Clinical cure	[27]
	51	M	0.41																					Longitudinal	DE and C	-	-	ITC	Improve	[28]
	61	F	0.66																					Longitudinal	DE, B and C	-	*Candida* spp.	ITC	Improve	
	49	F	0.41																					Longitudinal	DE, B and C	-	*C. tropicalis*	TRB, ITC	Improve	
	47	F	2																					Total diffuse	DE and C	-	*C. tropicalis, A. niger*	TRB, ITC	Improve	
	43	F	0.58																					Total diffuse	DE and C	-	-	ITC	Improve	
	61	M	0.25																					Distal partial diffuse	DE and C	-	*C. parapsilosis*	ITC	Improve	
	60	F	-																					Longitudinal	DE	-	-	TRB	Improve	[29]
	60	M	-																					Total diffuse, longitudinal	DE, B, C and PCR	-	*Cladosporium* spp.	ITC, EFZ	Improve	[30]
	63	F	-																					Total diffuse	DE and D	Irregular Black pigmentation, pseudo-Hutchinson sign, subungual hyperkeratosis	-	-	Improve	[31]
	43	F	-																					Total diffuse	DE and D	Irregular Black pigmentation,, subungual hyperkeratosis	-	-	Improve	
	67	F	-																					Total diffuse	DE and D		-	-	Improve	
	55	F	0.08																					Longitudinal	DE, B and C	-	*Candida* spp.	AMO 5%	Improve	[32]
	57.2	10F 10M	1.1	50% 1° Toe, 15% thumbnail, 10% 2° toe, 11 right and 9 left																				35% distal diffuse, 30% longitudinal, 20% distal linear, 10% Total diffuse, 5% proximal diffuse	DE	90% yellow streaks, 70% scaling, 65% multicolored pigmentation,50% reverse triangle, 35% subungual hyperkeratosis,	-	-	-	[15]
Mexico	68	M	-	2° fingernail hand																				Longitudinal	DE and C	-	*F. pedrosoi*	-	-	[33]
	21	M	0.25																					Total diffuse, longitudinal	DE, B, C and D	Irregular black pigmentation in longitudinal striae	*A. niger*	TRB, BIF, Urea	Improve	[34]
	74	M	-																					Total diffuse	DE, B and C	-	*A. niger*	-	-	[35]
	42	M	1																					Total diffuse	DE, B and C	-	*A. alternata*	-	-	[36]
	48	M	0.25																					Total diffuse	DE, B and C	-	*Scytalidum* spp.	TRB, urea	Improve	[37]
Taiwan	39	M	0.08																					Total diffuse, longitudinal	C and D	Subungual hyperkeratosis, multicolored pigmentation	*T. rubrum*	ITC	Cure	[38]
Turkey	53	M	0.5																					Total diffuse	ED and C	-	*C. albicans*	ITC	Cure	[39]
	51	13F 20M	4.8	42 nails, 61.9% 1° toe																				78.5% homogenous, 19% granular, 9.5 both and 9.5 longitudinal	ED and D	47.6% yellow streaks, 14.2% jagged edge, 9.5% Pseudo-Hutchinson sign	-	-	-	[40]
	42	4F 10M	0.66 to 10	20 nails																				35% striata, 25% Distal partial diffuse, 20% Proximal partial diffuse, 10% Distal linear, 10% Total diffuse	DE, C and D	95% Multicolored pigmentation, 95% yellow streaks, 95% irregular matt black pigmentation, 35% transverse striae, 15% jagged edges, 10% reverse triangle	90% *T. rubrum* 5% *C. albicans* 5% Negative	-	-	[41]
United States of America	51	F	8																					Distal lineal	ED and C	-	*W. dermatitidis*	BFZ 1%	Cure	[42]
	9	F	2																					Longitudinal	B and C	-	*Cromoblastomicosis*	-	-	[43]
	59	M	1																					Longitudinal	B and C	-	*C, parapsilosis*	FLZ	Improve	[44]
	57	F	1																					Diffuse	B and D	Reticular pigmentation, scaling, onycholysis	-	-	-	[45]
	67	M	0.75																					Total diffuse	C and PCR	-	*Sporothrix* spp.	ITZ	Cure	[46]
	79	M	0.33																					Longitudinal	C	-	*C. parapsilosis*	CLT	Improve	[32]

DE—Direct exam, B—Biopsy, C—Culture, D—Dermoscopy, PCR—Polymerase chain reaction. FLC—Fluconazole, TRB—Terbinafine, AMO—Amorolfin, RVZ—Ravuconazole, GRIS—Griseofulvine, EFZ—Efinaconazole, BIF—Bifonazole, CLT—Clotrimazole.

**Table 2 microorganisms-12-01096-t002:** Summary of the key points studied.

Classification	Dermoscopy Findings	Most Common Causal Agents	Therapy	Reference
More than 1 type 41% (n = 55) Longitudinal 21% (n = 28) Total diffuse 13% (n = 18) Distal diffuse 10% (n = 13) Distal linear 5% (n = 6) Proximal diffuse 2% (n = 2)	Multicolor pigmentation 31% (n = 46) Yellow-white spots 22% (n = 33) Superficial white scale 19% (n = 28) Hyperkeratosis 18% (n = 27) Inverse triangle pattern 15% (n = 22)	*Trichophyton rubrum* 55% (n = 39) *Neoscytalidium dimidiatum* 8% (n = 6) *Candida parapsilosis* 7% (n = 5) *C. albicans* 3% (n = 3) *C. tropicalis* 3% (n = 2)	Itraconazole 7% (n = 9) *Pulse treatment*: 400 mg daily (two 100 mg capsules twice a day for 1 week, and then three-week interval = 1 pulse; three pulses (1 week itraconazole + 3-week interval) or 3 months for toenail infection. Shorter for fingernail onychomycosis, possibly only 2 pulses. *Continuous dosing*: Conventional itraconazole 200 mg (2 hard capsules) once a day for 3 months, shorter for fingernail infection. Terbinafine 2% (n = 3) 250 mg once daily *Toenail infection*: 12 weeks *Fingernail infection only*: 6 weeks	[8,14,15,16,17,18,19,20,21,22,23,24,25,26,27,28,29,30,31,32,33,34,35,36,37,38,39,40,41,42,43,44,45,46]

## Data Availability

No new data were created or analyzed in this study. Data sharing is not applicable to this article.

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
