# Peer review of "Fungal Melanonychia: A Systematic Review"

_microorganisms, 2024, doi:10.3390/microorganisms12061096_

Round 1

Reviewer 1 Report

Comments and Suggestions for Authors

Fungal melanonychia can be caused by dematiaceous and non-dematiaceous fungi; the most common are Trichophyton rubrum and Scytalidium dimidiatum, followed by Alternaria and Exophiala. It is commonly seen in men with frequent toenail involvement. Fungi produce soluble, non-granular melanin that is incorporated into the nail plate. The involved nail typically has brown to black bands, best visualized with dermoscopy, and subungual hyperkeratosis with or without periungual inflammation. The pattern of nail involvement may indicate the causative agent. Longitudinal melanonychia is more common in dermatophytes such as Trichophyton rubrum, while diffuse pigmentation is observed in fungi such as Scytalidium, Aspergillus niger.

This review submitted by Carmen Rodríguez-Cerdeira and colleagues describes Fungal melanonychia as an uncommon condition, generally caused by melanin-producing pigmented filamentous fungi.

The authors also analyzed the possible clinical characteristics of patients diagnosed with fungal melanonychia through a systematic review of cases reported in the literature.

The authors ruled out the use of inappropriate articles based on the exclusion criteria; 33 articles with 133 cases were analyzed. 44% were women, 56% were men, the age range was between 9 and 87 years old.

the authors commented that most cases were reported in Türkiye, followed by Korea and Italy. The most frequent causal agents detected were Trichophyton rubrum and Neoscytalidium. Predisposing factors included nail trauma, migration history, employment, and/or outdoor activities. 45% of cases showed involvement of a single nail, while in 21% more than one affected nail was identified, ranging from 2 to 10 nails. Regarding clinical classification, 41% had more than one type of melanonychia, 21% corresponded to the longitudinal pattern and 13% were the total diffuse type. Likewise, the usual dermoscopic pattern was multicolored pigmentation. Based on the information collected, the authors described that fungal melanonychia is an uncommon variant of onychomycosis, the differential diagnosis is broad, which highlights the complexity of this disease.

Author Response

We are thankful to the referee for carefully reviewing the manuscript and the opinions regarding science and its presentation. We appreciate your positive reception for our efforts to provide a concise and clear summary of our work.

Reviewer 2 Report

Comments and Suggestions for Authors

Interesting topic to be reviewed. Well designed, well written. Congratulation. It is a very good paper. 

Author Response

We are thankful to the referee for tenderly reviewing the manuscript and the opinions regarding science and its presentation. Your support and praise motivate us to continue striving to contribute to the advancement of knowledge in our area of study.

Reviewer 3 Report

Comments and Suggestions for Authors

This is an interesting review about fungal melanonychia. The authors found a higher incidence of fungal melanonychia in Turkey, Korea and Italy and collected some useful information about dermatoscopy, clinical and locations of lesions.

In the following lines you will find some revisions to improve the article.

1.       Authors should add a paragraph about differential diagnosis among fungal melanonychia and other clinical similar conditions (e.g. melanoma, melanosis).

2.       I suggest to add a table to reassume the main results of the review. For example, I suggest to add a table with classification, dermatoscopy featurings, causal agents and therapies.

3.       I suggest to add an article about the importance of molecular biology in the diagnosis of mycosis and o identification the aetiology of fungal melanonychia. (doi: 10.1111/myc.13405).

Comments on the Quality of English Language

Good quality of English.
